# Prevention and Treatment of Sarcopenic Obesity in Women

**DOI:** 10.3390/nu11061302

**Published:** 2019-06-08

**Authors:** Maria L. Petroni, Maria T. Caletti, Riccardo Dalle Grave, Alberto Bazzocchi, Maria P. Aparisi Gómez, Giulio Marchesini

**Affiliations:** 1Unit of Metabolic Diseases and Clinical Dietetics, Sant’Orsola-Malpighi Hospital, “Alma Mater” University, via G. Massarenti 9, 40138 Bologna, Italy; maria.caletti@studio.unibo.it (M.T.C.); giulio.marchesini@unibo.it (G.M.); 2Department of Eating and Weight Disorders, Villa Garda Hospital, via Monte Baldo 89, 37016 Garda (VR), Italy; rdalleg@tin.it; 3Diagnostic and Interventional Radiology, IRCCS Istituto Ortopedico Rizzoli, via G.C. Pupilli 1, 40136 Bologna, Italy; abazzo@inwind.it; 4Department of Radiology, Auckland City Hospital, Park Road, Grafton, 1023 Auckland, New Zealand; pilucaparisi193@gmail.com

**Keywords:** sarcopenic obesity, aging, hormone replacement treatment, phytoestrogens, nutrition, exercise, physical therapy, body composition

## Abstract

Sarcopenic obesity (SO) is referred to as the combination of obesity with low skeletal muscle mass and function. However, its definition and diagnosis is debated. SO represents a sizable risk factor for the development of disability, possibly with a worse prognosis in women. The present narrative review summarizes the current evidence on pharmacological, nutrition and exercise strategies on the prevention and/or treatment of SO in middle-aged and older-aged women. A literature search was carried out in Medline and Google Scholar between 29th January and 14th March 2019. Only controlled intervention studies on mid-age and older women whose focus was on the prevention and/or treatment of sarcopenia associated with obesity were included. Resistance training (RT) appears effective in the prevention of all components of SO in women, resulting in significant improvements in muscular mass, strength, and functional capacity plus loss of fat mass, especially when coupled with hypocaloric diets containing at least 0.8 g/kg body weight protein. Correction of vitamin D deficit has a favorable effect on muscle mass. Treatment of SO already established is yet unsatisfactory, although intense and prolonged RT, diets with higher (1.2 g/kg body weight) protein content, and soy isoflavones all look promising. However, further confirmatory research and trials combining different approaches are required.

## 1. Introduction

In Europe, the prevalence of obesity in older adults has already reached epidemic proportions. In 2013, 19.9% of European women ≥ 50 years were affected by obesity, with a peak prevalence (21.6%) between 70 and 79 years [1]. In other non-European countries, obesity prevalence rates >20% in middle-age and elderly women have been reported [2]. Obesity in the elderly is associated with more advanced clinical disease stages and may in fact result in a significant number of years spent in chronic poor health. 

The term ‘sarcopenic obesity’ (SO) has been proposed to identify obesity with low skeletal muscle function and mass [3]. The concept stems from the study of sarcopenia in the geriatric population, since aging is accompanied by alterations in body composition. SO may lead to frailty, disability, and increased morbidity and mortality, which represent a significant burden on the health and social insurance systems. 

Many uncertainties still surround the condition of SO in terms of its definition, adverse short- and long-term health effect and clinical management [4]. As a matter of fact, studies on SO prevention and treatment are widely heterogeneous in terms of the definition of SO and methodologies employed for diagnosis, study design and outcome measures. A recently published systematic review on the effect of exercise alone or combined with dietary supplements included eight randomized controlled trials studies for a total of 604 patients [5]. As a consequence of the diversity of the methodologies employed and of the results observed, no clear conclusion or recommendation could be inferred. Alternatively, narrative reviews address a specific topic; the recent narrative review by Trouwborst and coll. [6] focused on nutrition and physical activity interventions in the prevention and/or treatment of SO, and the authors concluded that a combination of a moderate weight loss diet with concurrent exercise and a relatively high protein intake was able to ameliorate some parameters of SO. This review did not specifically report results about the impact of gender and did not include pharmacological treatment. 

The purpose of the present narrative review was to identify and summarize all that has been published so far about the prevention and/or treatment of SO limited to middle-aged and more mature women and to highlight new research areas not addressed so far.

### 1.1. Age-Related and Obesity-Related Changes in Muscle Composition, Structure and Function in Women

There is some controversy about the time of onset of age-related changes in fat-free mass—composed mostly of skeletal muscle—in women. Some authors [7] showed that body fat-free mass, measured by bioelectric impedance analysis (BIA), starts to decrease from 45 years onwards; for others [8], the decline in lean mass measured by dual-energy x-ray absorptiometry (DEXA) starts from age 58. Data from the NHANES cohort showed that women of European American and African American descent lose less than 1% total fat-free mass—measured by DEXA—during menopause but this figure decreases to −12 and −9% respectively between the age group of 40–49 and >75 years [9]. 

While in men hormonal changes have a pivotal role in the reduction of muscle mass, cross-sectional studies do not fully support the hypothesis that sarcopenia is mainly linked to estrogen deficiency in women, as is osteoporosis [10]. Parallel to changes in fat-free mass with aging, there is also a redistribution of fat mass mainly in the visceral component, but fat deposits are also observed in skeletal muscles and in the liver. Primary metabolic abnormalities have been described such as systemic and muscle oxidative stress, inflammation and insulin resistance, and adipose tissue derangement due to increased lipid storage. These alterations—which are interrelated—promote catabolic processes as well as a state of “anabolic resistance” to nutrients in the skeletal muscle [11]. Metabolic lipotoxicity secondary to ectopic fat accumulation in muscle tissue, mitochondrial dysfunction and muscle stem cell dysfunction with trans-differentiation into adipose cells have also been described [12]. Decreased resting metabolic rate as a consequence of loss of metabolically active fat-free mass, reduced physical activity and increased sedentary time all contribute to the development of obesity in women from mid- to old-age. 

Ageing in general is associated with lower muscle volume, decreased muscle fascicle pennation angle, decreased isometric and concentric contractile function but maintenance of eccentric function [13]. Middle-aged women with obesity (41–65 years) were noted to have a significantly lower peak knee extensor isokinetic torque than their younger counterparts (18–40 years) [14]. Elderly women with obesity have been found to have a larger lower limb muscle size and increased pennation angle [15,16]. Also, they have greater absolute maximum muscle strength compared to non-obese persons of same age [16]. However, they develop a lower force per unit of skeletal muscle than their normal-weight counterparts [13,16,17] and have a greater fat content in muscle. The likely explanation is that increased adiposity and body mass on one side load the antigravity muscles limbs—increasing muscle size and strength similarly to resistance training—but at the same time result in unfavorable muscle composition and architecture.

Indeed, most changes in muscle function associated with ageing and obesity are similar, since obesity can result in a phenotype typical of ageing even in relatively young individuals. Obesity and ageing have common mechanistic determinants such as chronic inflammation, decreased muscle protein synthesis and innervation, and impaired intramyocellular calcium metabolism. This explains why obesity-related changes may exacerbate the physiological muscle ageing process [13]. Further research is needed in order to understand the effects of obesity on skeletal muscle ageing. 

### 1.2. Risk Factors for Sarcopenic Obesity and Related Disability

Excess weight burden leads to a vicious circle causing the reduction of physical activity, osteoarthritis and accretion of adipose tissue as well as the deterioration of muscle mass and function. SO has been shown to precede the onset of instrumental activities of daily living (IADL) disability in the community-dwelling elderly with a risk approximately 2.5 times higher than in individuals with non-SO [10]. The impact of SO on disability in different sexes has not been fully elucidated; some studies showed no difference in incidence [10], others showed a worse prognosis in women [18]. 

A number of risk factors for the development of SO have been highlighted. A putative role for the polymorphisms of TP53 Arg/Arg and for 308 G/A TNF-α in sarcopenia and in SO has been proposed [19,20]. Also, low vitamin D status exerts a detrimental effect on muscle function, while there is evidence for a beneficial effect of vitamin D supplementation on muscle strength, physical performance and the prevention of falls in the elderly female population [21]. Obese individuals are often deficient in vitamin D, especially women due to their relative larger adipose tissue mass than their male counterparts. Epidemiological data suggest a role for vitamin D deficit in SO development [22]. Weight loss (intentional and non-intentional) and weight cycling represent other potential risk factors. With weight change in old age, significantly more lean mass is lost with weight loss than is built up with weight gain [23]. This suggests that weight loss and weight cycling could accelerate sarcopenia in older women with overweight/obesity as well as in men [24]. As a consequence, strategies to counteract loss of muscle mass during weight loss have been advocated. 

## 2. Methods of Narrative Review

A literature search was carried out in Medline and Google Scholar in order to identify relevant articles. The search was carried out between 29th January and 14th March 2019. English language papers were included if they were published in a peer-reviewed journal. 

To start with, the following search strategy was used: (“sarcopenic obesity“) OR (“sarcopenia” AND “obesity”) AND (drug* OR pharmacological OR hormone replacement therapy OR supplement* OR amino acid* OR diet OR nutrition OR nutraceutical* OR protein OR vitamin OR mineral OR exercise OR physical activity OR gait speed OR walking speed OR handgrip strength OR strength). Additionally, the following keywords were used: “energy restriction OR weight loss” AND “skeletal muscle OR body composition”. The limitations “human”, “female” and “middle-aged AND aged (>45 year)” were applied to the search parameters. Further publications of potential interest were identified as citations in the articles retrieved during the first search. Only controlled intervention studies were included. Those involving procedural therapies (endoscopic treatments or bariatric surgery) or carried out in subjects who had been treated for oncological conditions over the previous 12 months were excluded because of potential confounders. Acute and short-term (i.e., ≤1 week) treatments were also excluded. 

The search strategy was further refined by including only intervention studies in which the focus was on the prevention and/or treatment of sarcopenia (or the prevention of fat-free mass loss during intentional weight reduction) associated with obesity and/or with SO. All selected studies could be retrieved as full papers. Since the search targeted mid-age and older women, investigations involving subjects younger than 45 without a separate analysis for age groups were excluded, as were those that enrolled men only. Studies incorporating both male and female populations were included only if there was a predominant (subjectively defined as ≥ 80% enrolled subjects) presence of women—alternatively, if a separated analysis for differences between sexes had at least partly been carried out. Studies that enrolled overweight subjects (defined as BMI between 25 and 30 kg/m^2^) together with obese subjects were also included. The publication date of the retrieved studies ranges between 2001 and 2019. The review comprises 24 papers including 1820 women (90%) out of a total number of 2014 enrolled subjects.

## 3. Definition of Sarcopenic Obesity 

The identification of SO is a currently debated issue, since BMI and waist circumference are inadequate to evaluate muscle mass loss in the elderly population. SO is currently defined as the combination of sarcopenia (see below) and obesity, the latter defined as a body mass index (BMI) ≥ 30 kg/m^2^ (in certain ethnic groups ≥ 27.5 kg/m^2^). BMI is not useful for identifying the status of obesity or as an outcome measure and should be abandoned as it inaccurate [25]. Alternatively, obesity could be diagnosed by cutoffs of percent body fat or other adiposity indices. Regrettably, once more, there is heterogeneity about the level of fat mass to be used as a cut-off [26], since most published values range between 30 and 40% or even higher. In 2015, fat mass ≥ 32% in women (measured by DEXA) has been proposed as a consensus cut-off [27]. A combination of body mass index and adiposity measures, i.e., fat mass index (FMI) and fat-free mass index (FFMI), has also been reported in epidemiological studies [25].

A number of methods for evaluating body composition, based on the assessment of both adiposity and muscle mass, are currently being used. Computed tomography (CT) and magnetic resonance imaging (MRI) represent the gold standard for estimating total and segmental fat mass (especially visceral fat), as well as muscle mass (cross-sectional area and volume) in the research setting [28]. They additionally allow the evaluation of muscle density (which relates to intramyocellular lipid deposits) as well as intramuscular and subcutaneous adipose tissue accumulation. Air displacement plethysmography measures body volume and body density, providing a non-invasive estimation of total lean mass and fat mass and can equally be applied to patients with morbid obesity [29]. DEXA provide estimates of the lean and fat mass of the entire body or in specific body regions, e.g., the appendicular region. It is relatively inexpensive, and it also provides the advantage of estimating bone mass and density, thus allowing the diagnosis of the triad of bone, muscle, and adipose tissue impairment, i.e., osteosarcopenic obesity [30]. 

However, in individuals with overweight/obesity, the appendicular skeletal muscle mass (ASM) either expressed as centile or normalized by the square of the height (h^2^) can underestimate sarcopenia [25,30], and other criteria, e.g., ASM adjusted for total fat mass or adjusted for height and body fat mass (residuals method), have been proposed [31,32]. Bioelectrical impedance analysis (BIA) is a simple and low-cost technique, but its estimation of body composition is indirect, through measurement of whole body and segmental reactance and resistance affected by fluid retention and disease-related conditions. For these reasons, despite its widespread use also in clinical trials, the use of BIA as an assessment tool of muscle mass for diagnosing sarcopenia has been unrecommended in a consensus statement [33]. A further level of complexity is represented by infiltrated fat in muscle and bone, which contributes to limb adiposity, but it can be concealed and therefore hard to detect [25]. 

While the assessment of body composition is adequate for the diagnosis of obesity, this does not suffice for the diagnosis of sarcopenia according to a forthcoming evidence-based definition of sarcopenia [34]. Unlike diagnosis based on DEXA alone, the diagnosis of sarcopenia based on grip strength is associated with mortality, hip fracture, falls, mobility disability and IADL disability. Cut-off points in grip strength or grip strength/BMI are the best tools to identify women (and men) at risk for mobility disability. 

A clinical diagnosis of “sarcopenia” might also include functional limitations such as slow walking, difficulty in rising from a chair without hands or walking up stairs. This is especially relevant for decision-making about interventions other than physical activity or other lifestyle changes [34]. 

As a consequence, the definition of sarcopenia in women according to the European Working Group on Sarcopenia in Older People (EWGSOP1) [35] has recently been updated (EWGSOP2) and it is based on three criteria: 1) low muscle strength; 2) low muscle quantity or quality; 3) low physical performance [36]. Low muscle strength is defined as grip strength below 16 kg (27 kg in males) and/or chair stand >15 s for five rises. Low muscle quantity or quality is defined as appendicular skeletal muscle mass (ASM) < 15 kg (20 kg in males) or ASM/height^2^ less than 6.0 kg/m^2^ (7.0 kg/m^2^ in males). The cutoffs for low physical performance are: -Gait speed ≤ 0.8 m/s-Short Physical Performance Battery (SPPB) ≤ 8-point score-Timed-Up and Go test (TUG) ≥ 20 s-Non-completion or ≥ 6 min for completion of the 400-m walk test.

Criterion 1 identifies probable sarcopenia. Diagnosis is confirmed by additional documentation of Criterion 2. If all three criteria are met, sarcopenia is considered severe. Note that these criteria are different from those identified in 2010 (EWGSOP1), i.e., skeletal muscle index less than 6.76 kg/m^2^; gait speed less than 1 m/s; grip strength below 20 kg [35], and those which have been used in some intervention studies reported in the present review. 

The EWGSOP2 document, however, remarks that sarcopenic obesity represents a distinct condition [36] and the lack of consensus on its definition and diagnosis represents a recognized limitation requiring widespread coordinated action among researchers and clinicians [12]. In a recent paper from El Ghoch and coll., the six-minute walking test was the only independent test associated with low lean body mass, but the 4-m gait-speed test was shown to represent an accurate functional test for SO screening in female patients [37]. 

## 4. Prevention of Sarcopenic Obesity 

Prevention of SO can be defined in terms of interventions aimed at preserving skeletal muscle function and mass in obesity [12]. It is, however, difficult to entirely differentiate between those interventions aimed at prevention and those aimed at the treatment of SO in mid-age and old-age women. This is because in most research articles, a clear-cut differentiation of sarcopenic from non-SO is missing and often only indirect markers (e.g., physical independency) are provided to exclude overt severe sarcopenia. Other studies have explicitly enrolled women with a condition of pre-sarcopenia or with some functional impairment. In a few cases, non-sarcopenic and sarcopenic women with obesity have been enrolled in intervention protocols and pooled results have been reported. When SO is present, it is questionable whether intervention aimed at preserving muscle mass while inducing loss of fat mass represent secondary prevention (i.e., aimed at reducing the impact of the already present disease) or treatment itself.

Intervention strategies have either been nutritional or pharmacological or exercise-based or a combination of the above. For the purpose of clarity, interventions based on a single strategy are separated from those combining two or more intervention regimens.

### 4.1. Single Interventions

#### 4.1.1. Nutrition

Two studies—from the same research group—have investigated the effect of nutritional strategies alone for the prevention of weight loss-related sarcopenia in women with obesity (Table 1.) Porter Starr et al. [38] studied the effect of a 6-month moderately hypocaloric diet (~500 kcal energy deficit) with either normal protein (0.8 g/kg) or high protein (1.2 g/kg) content in the frail elderly—mean age 68 years, mainly women—with obesity and functional impairment. Both interventions reduced body weight by approximately 8% on average and improved handgrip strength and short physical performance battery (SPPB) score as compared to baseline; notably, the amelioration of SPPB in the high protein group was greater than in the normal protein group. However, in a subsequent trial carried out in a younger (mean age 60 years) all-female population, both weight reduction diets (average weight loss, 6%) proved to be safe and effective in improving physical function with no added benefit from a higher protein content in the diet [39]. Moreover, in these studies, a modest but significant loss of lean mass (between –10% and –24% of total mass loss) was observed with both diets, with a non-significant trend for a lower reduction in the high protein group. A noteworthy observation is that in the latter study, black elderly females lost less body weight and experienced lower improvement in the 6-min walking test (6MWT) than white participants [39], confirming previous reports in the literature suggesting a reduced effectiveness of weight loss interventions in black women [40,41].

#### 4.1.2. Pharmacotherapy

Two studies in which pharmacological intervention was carried out as the sole treatment for the prevention of SO have been identified (Table 2). The effect of hormone replacement therapy (HRT) on body composition in post-menopausal women has been compared versus placebo in a small cross-over study on 16 subjects [42]. Despite the short-term intervention (12 weeks), HRT—aside from the favorable well-known effect on bone density—was apparently not only able to prevent the loss of lean body mass occurring during placebo intake, but also to increase it in absolute terms. At the same time, HRT decreased abdominal fat mass, while total body weight was unchanged. 

In the second study, the effect of the supplementation of vitamin D (cholecalciferol 10.000 UI or placebo three times a week) was tested in a gender-mixed population of pre-sarcopenic subjects (obese and non-obese) with vitamin D deficit. Cholecalciferol administration had no effect on handgrip strength. However, it was associated with increased appendicular skeletal muscle mass (ASMM). When data were analyzed to account for subgroups and the interaction between vitamin D and obesity on ASMM, the effect size of vitamin D on muscle mass was much higher in subjects without vs. subjects with obesity. Unlike non-obese subjects in whom a higher percent change in ASMM was observed in males compared to females, no sex-related effect was observed in the group with pre-sarcopenic obesity [43].

#### 4.1.3. Exercise

Two studies from Brazil met the inclusion criteria on the effect of exercise alone for the prevention of SO (Table 3). Cunha et al. studied the effect of resistance training (RT) three times/week at two different levels (1 or 3 sets of 10–15 repetitions maximum for each exercise, i.e., 30- and 50-min duration, respectively) vs. controls (no exercise) on components of SO and on bone density in a sample of non-disabled women aged 60 and over. Both RT strategies similarly increased skeletal muscle mass vs. controls, but the 50-min sessions resulted in a significantly higher increase in strength and a modestly greater reduction in fat mass than the 30-min sessions [44]. Beneficial effects of RT were found in the study by de Oliveira Silva et al. comparing non-sarcopenic with sarcopenic women with obesity; in the non-sarcopenic subgroup RT resulted in the amelioration of functional tests and strength as well as a reduction in fat mass compared to pre-training values [45]. Two additional studies evaluating aerobic exercise (AE) as the sole intervention fulfilled the inclusion criteria for the present review. In the study by Davidson et al. AE was compared with RT alone and with the combination of both [46]. This study included both men and women, who were analyzed separately, but unfortunately only pooled results were presented, since no sex-related differences were found. The authors conclude that AE improved tests of functional limitation in similar fashion to RT alone but combined AE + RT was superior to both. Increased skeletal muscle mass was only observed following both RT and RT+AE and an improvement of cardiorespiratory fitness only occurred following AE. The other study investigated AE and hypocaloric diet in the framework of a weight loss intervention. However, one of the trial arms consisted of an AE intervention without diet and will be reported in paragraph 4.2.1 [47].

### 4.2. Combined Interventions

Five studies of combined interventions for the prevention of SO were identified. Of these, four included a weight loss intervention, while one did not (Table 4).

#### 4.2.1. Exercise Plus Nutritional Therapy

The largest prevention study on weight loss so far carried out enrolled overweight or obese postmenopausal, mainly (83%) non-sarcopenic, sedentary women. They were randomized to dietary modification (goals of 1200–2000 kcal/day and 10% loss of baseline weight within six months followed by weight maintenance), to moderate-to-intense AE (AE-5 sessions/week for a total of 225 min/week), to diet + exercise or to no intervention (control group) for a 12-month period [47]. At 12 months, the weight changes averaged −2.4% (*p* = 0.03) in the exercise group, −8.5% (*p* = 0.001) in the diet group, and −10.8% (*p* = 0.001) in diet + exercise group, compared with −0.8% among controls. Hypocaloric diet alone resulted in a significant loss of both total (−1.1 kg, *p* < 0.001) and appendicular (−0.5 kg, *p* = 0.02) fat-free mass. Both measures of fat-free mass remained unchanged over the limited weight loss in the AE alone group. Despite the largest weight loss occurring in the AE + diet group, there were only modest losses in total (−0.4 kg) and appendicular fat-free mass (−0.2 kg), and significantly lower losses (*p* < 0.01) than in the diet-only group. However, in women who were not sarcopenic at baseline, no between-group differences in the incidence of sarcopenia were found at 12 months (between 7 and 10%).

Another study evaluated the effects of whey protein (25 g t.i.d.) or isoenergetic carbohydrate supplements to hypocaloric diets coupled with mild exercise (flexibility and aerobic 40′–50′ sessions 2–3/week) [48]. Whey protein supplementation resulted in a borderline significant (*p* = 0.051) greater weight loss but also in a significantly greater absolute fat-free mass loss. However, after correction for weight loss, the relative muscle volume showed a greater net gain of muscle in the protein-supplemented diet as compared to the carbohydrate-supplement group (*p* = 0.049). A greater loss of intramuscular adipose tissue (*p* = 0.03) was also observed. However, no differences in changes in strength, balance, or physical performance measures were found between diets, possibly because of the limited sample size.

Galbreath et al. studied the effect of a 6-month RT (3 times/week) intervention coupled with a moderately hypocaloric diet (1st week 1200 kcal, 2nd to 12th week 1600 kcal/day) either normal protein (0.8 g/kg) or high protein (1.2 g/kg). A third group had RT only. All three groups underwent significant improvements in muscular strength, muscular endurance, aerobic performance, balance and functional capacity [49].

#### 4.2.2. Exercise Plus Pharmacotherapy

The prevention of changes in body composition following menopause was the scope of a trial aimed to study the interaction between hormone replacement therapy (HRT) and exercise training. Post-menopausal women aged 40–65 were randomized to exercise (resistance + weight bearing training) or to no-exercise according to their HRT status [50]. Exercise was reported to provide a significant improvement in body composition (increase of total and appendicular fat-free mass, decrease of body fat) and strength, as compared to no-exercise; however, in the exercise groups, women who were already on HRT did not gain further benefit as compared to women who were not on HRT.

Resistance training coupled with the PPARγ-agonist pioglitazione—an antidiabetic medication shown to reduce abdominal visceral adipose tissue—was tested on body composition changes in non-diabetic elderly overweight/obese of both sexes at risk of mobility disability [51]. During a weight reduction diet (15% protein, ~500 kcal energy deficit), women who received RT (with or without pioglitazione) lost less thigh muscle volume than those who did not received RT. To be noted, in women—unlike in men—pioglitazone not only did not contribute to visceral fat loss but also contrasted with the loss of subcutaneous fat.

## 5. Treatment of Sarcopenic Obesity

The definition of the treatment targets in SO is yet unclear. The joint statement of the European Society for Clinical Nutrition and Metabolism (ESPEN) and European Association for the Study of Obesity (EASO) focuses on improvement of skeletal muscle function and mass [12]. Both reduced burden of disabilities and comorbidities, as well as improved quality of life represent treatment targets far more important from patients’ perspectives. Unfortunately, very few data in this area have been published so far. This issue is particularly critical in elderly women, at high risk of SO per se, where any effort aimed at weight loss may produce untoward, negative effects [52]. Strategies aiming at preserving (rather than increasing) muscle mass during weight loss in clearly defined SO have also been reported in the present section. Most intervention studies in women include exercise training or physical therapy. However, nutritional and pharmacological strategies have also been studied, as single treatment modality or in various combinations.

### 5.1. Single Treatment

#### 5.1.1. Nutrition

Applicable studies are summarized in Table 5. Adding a high-quality protein food (210 g of ricotta cheese daily for 3 months) to habitual diet did not improve appendicular skeletal mass or strength in both women and men with sarcopenia, most of whom with concurrent obesity; interestingly, more favorable trends were observed for men [53]. The impact of proteins on the preservation of skeletal mass and function during weight loss in women has been studied in two separate trials. Muscariello et al. [54] reported that a high protein (1.2 g/kg) hypocaloric diet over three months resulted in the preservation of arm muscle mass, while a normal protein diet produced a modest but significant loss (−5.7 cm^2^, *p* < 0.001 vs. baseline). Muscle mass index—as measured by BIA—significantly increased (*p* < 0.01 vs. baseline) following the high protein but decreased following the normal protein diet (*p* < 0.01 vs. baseline). Handgrip strength was maintained with both diets. Sammarco et al. confirmed the favorable effect of high-protein hypocaloric diet by the use of BIA [55]. In this pilot trial, women with SO on a hypocaloric diet were randomized either to protein supplementation (to reach 1.2–1.4 g/kg) or to placebo. Women in the placebo group showed higher loss of lean body mass compared to those in the protein-enriched diet group (−1.3 kg vs. −0.5 kg; *p* < 0.05). Handgrip strength improved in the high protein diet group (+1.6 kg; *p* = 0.01 vs. baseline), while it was unchanged in the placebo group. The general health domain of quality of life (Short Form-36 Questionnaire) also improved significantly in the high protein group, while no change was observed for other categories or for the score of SPPB.

#### 5.1.2. Pharmacotherapy

The effect of a 6-month administration of soy isoflavones as compared to placebo on changes in body composition was assessed in a sample of 18 post-menopausal women with SO [56]. Isoflavones proved significantly superior to placebo in increasing leg (*p* = 0.016) and appendicular (*p* = 0.034) lean mass, as well as muscle mass index (*p* = 0.037) (Table 6).

#### 5.1.3. Exercise and Physical Therapy

Seven studies that used exercise (RT alone or associated with aerobic training) as the sole treatment modality in women with SO were identified (Table 7). Among studies with a no-exercise arm for comparison, Gadelha et al. showed that RT (3/week over 24 weeks) improved strength-related variables (*p* < 0.001 vs. baseline and vs. controls), appendicular FFM (+0.29 kg; *p* < 0.001 vs. control), and total FFM mass (+0.6 kg; *p* < 0.01 vs. baseline and vs. controls) [57].

The differential and additive effect of RT and AE (2/week over 8 weeks) was studied in a population with a large majority of women (83%) [58]. All exercise groups (RT alone, AE alone, RT + AE) improved back extensor strength vs. controls. Only RT as the sole treatment modality increased handgrip strength (+3.5 kg; *p* < 0.05 vs. all other groups). Similarly, improvement in strength, not in functional performance, was found in the small subgroup of women with SO in the RT treatment arm, in the study by de Olivera Silva et al., who also had a low frequency of exercise (2 session/week) [45]. On the contrary, Park et al. [59] found that combined RT and AE training (5/week for 24 weeks) resulted in increased handgrip strength (+2.5 kg; *p* < 0.001 vs. baseline and vs. control) and walking speed (+0.15 m/s; *p* < 0.01 vs. baseline and vs. control). A reduction in fat mass (−2.0 kg; *p* < 0.01 vs. control) was also observed, with no effect on appendicular lean mass.

Two studies evaluated elastic band RT, an exercise modality that is simple, inexpensive and can be carried out at home without the need of attending a gym. Liao et al. carried out a 12-week intervention study followed by a further follow up at 6 months after the end of rehabilitation intervention [60]. Elastic band RT proved effective on all sarcopenia components (muscle mass measured by BIA, strength, mobility) and also in improving quality of life. Moreover, the effects of RT were clinically significant and sustained over time at 9-month follow-up: in RT vs. controls, there was an increase in absolute muscle mass (+0.72 kg; *p* < 0.01), in global physical capacity score (+4.22; *p* < 0.001), in the physical component score of the short-form 36 questionnaire (SF-36) (+15.06; *p* < 0.001). These results were not confirmed in a study on the effects of elastic band exercise training (3/week) as compared to standard home exercise on body composition measured by DEXA [61]. No effects of RT were demonstrated on appendicular lean mass, while fat mass was reduced, and bone density increased vs. controls. The reasons for non-univocal outcomes between the two studies—which were comparable in terms of mean age and BMI—could be ascribed to differences in control interventions (non-active vs. active control group), treatment protocols, and body composition assessment modalities (BIA vs. DEXA).

One study compared two modalities of RT for 15 weeks: standard strength hypertrophy training with high-speed power training circuit [62]. Only power training improved the physical function (SPPB) of 20% (*p* = 0.02 vs. baseline). However, exercise did not improve body composition, 6MWT and handgrip strength vs. baseline, and produced only negligible-to-small improvement in IADL tasks. The small sample size (eight subjects for each group) limits the significance of these negative results.

### 5.2. Combined Treatments

Three controlled studies have examined the combination of nutritional intervention and exercise and/or physical therapy in sarcopenic women (Table 8). In two of them, a hypocaloric diet was part of the treatment.

Kemmler et al. tested the effect of weekly sessions of whole-body electromyostimulation (WB-EMS) over 26 weeks with/without protein and vitamin D supplementation vs. with a non-training control group while on an isocaloric diet [63]. In both WB-EMS groups, an increase of skeletal muscle mass index was found: WB-EMS +0.14 kg/m^2^, WB-EMS plus protein +0.11 (both <0.001 vs. control). A marginal, although statistically significant, increase in gait speed was observed only in the WB-EMS group (+0.08 m/s; *p* = 0.026 vs. control). No significant changes in body fat or handgrip strength were demonstrated in all treatment groups. In a 4-arm RCT of 12-week duration, Kim et al. investigated the effect of twice weekly RT plus AE alone vs. nutritional intervention (3 g of essential amino acids plus catechins plus vitamin D supplementation) alone vs. combination of exercise and nutrition vs. control. Reduction in body fat mass was significant vs. control only in the combined treatment, −1.0 kg (*p* = 0.036). Both exercise groups increased step length vs. control. No significant changes in SMI and grip strength were found among groups [64].

Finally, in a subgroup of women with SO, in the study by Mason and coll. [47], the 12-month effect of intense (225 min/week) AE alone or combined with/without a moderately hypocaloric diet was compared with a control group of no intervention. At the end of the study, 14% of cases in the control group, 8% in the diet group, 50% in the exercise group, 35% in the diet + exercise group no longer met the criteria for sarcopenia by 12 months (no subgroup-specific statistical analysis was provided).

No studies combining RT with nutrition intervention have been reported at the time of the present review.

## 6. Discussion

Women with obesity and low muscle mass/function are at increased risk of frailty and disability. Metabolic and lifestyle abnormalities [65] compromise the ability to preserve muscle function and mass, especially when chronic diseases co-exist with obesity. Insulin resistance has been shown to contribute to muscle weakness and to the “dynapenic obesity” phenotype of middle-aged women with the metabolic syndrome [66]. Muscle fat accumulation linked to insulin resistance reduces muscle density and quality with lower contractile protein content per mass unit [66].

Weight loss regimens represent a further leading risk factor for the development or worsening of SO [67]. In a retrospective analysis of a caloric restriction and exercise weight loss intervention in postmenopausal women, Bopp et al. found that the average loss of lean mass was clinically significant, representing approximately one-third of the total mass lost at a protein intake of 15–20% of energy (approximately 0.62 g/kg body weight/day). Additionally, they found that participants lost 0.62 kg less lean mass for every 0.1 g/kg body weight/day increase in dietary protein beyond the standard [68].

Effective strategies are urgently required to reduce the burden of morbidity and mortality in a rapidly increasing obese population [12]. Nutritional, pharmacological and exercise/physical activity treatment are available to prevent SO as well as to reduce the burden of SO in post-menopausal and elderly women, and are summarized in Figure 1.

### 6.1. Prevention of Sarcopenic Obesity

Most RCTs in non-sarcopenic women with obesity have focused on the prevention of sarcopenia during hypocaloric diets for weight loss. Altogether, they suggest that hypocaloric diets (500 kcal deficit) with a protein content of at least 0.8 g/kg provide significant weight and fat loss at the same time as improving physical function. A higher (1.2 g/kg) protein diet is expected to generate possible added benefits in contrasting lean mass reduction. Adding AE to a normal-protein hypocaloric diet reduces lean mass wasting but does not totally prevent the development of sarcopenia. Whey protein supplements do not seem to provide benefit when coupled with mild exercise only. These findings might not be generalizable to mixed (female and male) populations. Backx et al. failed to demonstrate a significant effect of a very high protein diet (1.7 g/kg) on the prevention of lean mass loss in a mixed-gender elderly population [69]. On the contrary, Beavers et coll. found a sparing effect of high-protein diet on lean mass over 6 months in the elderly (74% women) with obesity, with no detrimental effect on gait speed [70].

The only intervention that was definitely effective in the prevention of all components of sarcopenia was the combination of RT (thrice weekly) with either a normal (0.8 g/kg) or high protein (1.2 g/kg) hypocaloric diet, leading to significant improvements in muscular strength, endurance, aerobic capacity, balance and functional capacity [49]. Also, in the absence of dietary intervention, RT reduces fat mass, increases muscle strength and improves functional capacity in women at risk of SO. On the contrary, AE alone ameliorates functional capacity, not strength or lean mass. Physical therapies (muscle electrostimulation and vibrations) are either ineffective or add little to exercise.

The results are confirmed in mixed-gender trials. Elderly subjects of both sexes with obesity and mild-to-moderate frailty lose less lean mass and gain more strength when RT alone [71] or RT + AE is added to a hypocaloric diet [60,62,72]. Nicklas et al. also found that RT coupled with a hypocaloric diet improved mobility and strength, despite the calorie restriction-related loss of lean mass [65]. Houston et al. recalled a random sample of 60 older adults (mean age at randomization, 67.3 years; 69% women) who had been randomized to caloric restriction plus exercise or exercise only in five RCTs on average 3.5 years before. They found that physical performance was similarly maintained in both exercise groups, whereas the favorable changes in weight and body composition were not [73].

In a small-scale study, hormone replacement therapy (HRT) showed some advantage in preventing the negative body composition changes of post-menopausal age, increasing lean body mass and decreasing abdominal fat mass, without changes in total body weight [42]. In another study using exercise as strategy, HRT did not gain further benefit as compared to women who were not on HRT [50]. Regrettably, no additional studies are available, possibly because of the concern about an increased risk of cancer linked to estrogens in obesity. Correction of vitamin D deficiency—which is often present in obesity—can produce beneficial effects on lean mass [43] and should routinely be carried out.

### 6.2. Treatment of Sarcopenic Obesity

In women with sarcopenic obesity, there is concordance among RCTs that a higher protein content in the diet may effectively contrast lean mass loss and may yield some benefit on strength during hypocaloric diets but cannot ameliorate SO during isocaloric diets.

On the whole, RT—delivered with different regimens and possibly combined with aerobic training—improves strength and physical function, especially if intense (≥ 3 sessions/week) and prolonged (≥ 3 months). A beneficial effect on muscle mass was not consistent across the studies. The scarcity and heterogeneity of RCTs prevent any firm conclusion about the efficacy of the combination of nutritional intervention (with either hypocaloric and isocaloric diet) and exercise on SO.

Only one RCT on pharmacological treatment used soy isoflavones in post-menopausal obese sarcopenic women [56] and showed a beneficial effect on muscle mass, but data require confirmative trials due to the limited number of patients.

### 6.3. Sex-Related Aspects

Sex hormones have pivotal roles in maintaining skeletal muscle homeostasis. Under normal conditions, the different roles of estrogens and androgens contribute to sex differences in skeletal muscle morphology and function. Testosterone is a powerful anabolic factor promoting protein synthesis and muscular regeneration, mainly via increased muscular expression of insulin-growth factor-1 (IGF-I). Estradiol reduces the progressive muscle atrophy in postmenopausal women, suggesting an anti-inflammatory and anti-catabolic influence of estrogens on skeletal muscle in women, especially after exercise. However, further research is awaited to support a significant effect of estrogens on muscle mass [74].

Oikawa and colleagues have recently highlighted sex-based differences in the ability to recover muscle strength in the elderly. Two weeks of combined calorie restriction (with maintenance of normal protein intake) plus reduced physical activity resulted in decreased muscle isometric strength in older men and women. However, upon resumption of physical activity and caloric intake, women did not recover strength as measured by maximum voluntary contraction, while men did [75]. In the male sex, single muscle fiber power and isometric tension were unchanged or paradoxically increased (muscle biopsies were not carried out in women). This finding suggests impaired resiliency, which could result in greater functional impairment over time. Whether estrogens as HRT or soy phytoestrogens may overcome this defect warrants assessment by future studies.

The present review does not provide a definite answer about which sex responds best to exercise and nutritional strategies targeting sarcopenic obesity. Only two studies were identified including a comparison between sexes. In the study by Davidson and coll. [46], skeletal muscle mass—measured by MRI—muscle strength measures and changes in functional limitations did not reveal any differences among gender following the exercise interventions. The sole exception was a greater reduction of total and visceral fat in men than women in the aerobic exercise group only. Similarly, a trend towards increased appendicular muscle mass measured by DEXA in men but not in women was found in the cheese protein supplementation study by Aleman-Mateo and coll. [53] which did not attain statistical significance possibly due to the small number of study subjects.

Pharmacological agents other than estrogen have demonstrated sex-specific effects on body and muscle composition. For example, pioglitazone resulted in significant sex-based differences in abdominal fat, where abdominal fat loss was significantly greater in men, with no change in women [51]. Vitamin D supplementation showed a gender effect by increasing ASMM only in presarcopenic non-obese male subjects, while no sex-related difference was found in obese subjects [43].

## 7. Conclusions

SO in women represents a condition under research scrutiny with regard to definition, diagnostic criteria and optimal treatment. At present, intense and prolonged RT has definite efficacy in the prevention and/or treatment of SO. Adequate protein content in the diet and correction of vitamin D deficiency are also required. This conclusion supports the ESPEN/EASO recommendation of coordinating action aimed at reaching consensus on optimal treatment with particular regard to nutritional therapy [12]. However, more research on optimal nutritional strategies in weight loss protocols and combined approaches is required.

## Figures and Tables

**Figure 1 nutrients-11-01302-f001:**
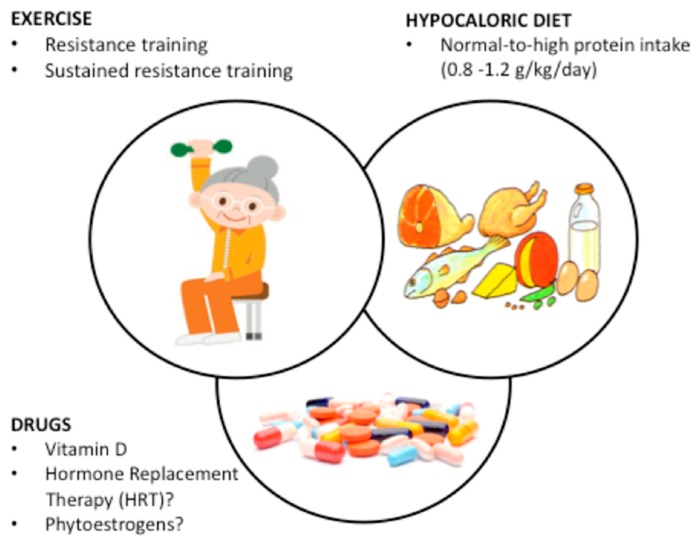
Overview of interventions for the prevention and/or treatment of sarcopenic obesity in women based on a non-systematic review of 24 papers including 1820 women. The three layers of intervention are superimposed to show that they are not mutually exclusive. Resistance training is the most effective strategy with effects directly related to its frequency and duration; when coupled with hypocaloric diets with normal-to-high protein content the effects on both prevention and treatment are amplified. Vitamin D deficit should be corrected whenever present. Evidence on HRT and phytoestrogens needs confirmation by future research.

**Table 1 nutrients-11-01302-t001:** Prevention of Sarcopenic Obesity—Nutritional Intervention.

Ref.	No. Subjects Age (years)	Inclusion Criteria	Design	Type of Intervention	Intervention Effect	Notes
[38]	67 (80% women) Mean age, 68	BMI ≥ 30 kg/m^2^, functionally impaired (SPPB score of 4–10 out of 12)	Parallel group (*n* = 2) RCT Duration 6 months	Normal protein (NP, 0.8 g/kg) or high protein (HP, 1.2 g/kg), moderately hypocaloric diet No exercise	WL (kg): −7.5 ± 6.2 (NP); −8.7 ± 7.4 (HP), both *p* < 0.001 vs. BL Lean mass (kg): −1.8 (NP) −1.1 (HP), both *p* < 0.01 vs. BL Total SPPB score: +0.9 (NP), *p* < 0.01 vs. BL; +2.4 (HP), *p* = 0.02 vs. NP, *p* < 0.001 vs. BL HGS (kg): +1.3 (NP) +1.1 (HP), both *p* < 0.01 vs. BL	Mean BMI = 37.1 kg/m^2^ Body composition measured by air displacement plethysmography
[39]	80 women ≥ 45 years (48.8% white) Mean age, 60	BMI ≥ 30 kg/m^2^	Parallel group (*n* = 2) RCT Duration 6 months	Normal protein (NP, 0.8 g/kg) or high protein (HP, 1.2 g/kg) moderately hypocaloric diet No exercise	WL (kg): −6.2 (NP), −6.4 (HP), both *p* < 0.001 vs. BL. WL greater in white (–7.2) than in black women (−4.0), *p* < 0.04 Lean mass (kg): −1.0 (NP) −0.6 (HP), both *p* < 0.01 vs. BL Total SPPB score: +1.2 (NP); +1.0 (HP), *p* < 0.001 vs. BL 6MWT (m): +46.8 (NP) +46.9 (HP), both *p* < 0.001 vs. BL	Mean BMI = 37.8 kg/m^2^ Body composition measured by air displacement plethysmography

AE—aerobic exercise; BL—baseline; BMI—body mass index; HGS—handgrip strength; 6MWT—6-min walking test; PL—placebo; RCT—randomized controlled trial; SPPB—short physical performance battery; WL—weight loss.

**Table 2 nutrients-11-01302-t002:** Prevention of Sarcopenic Obesity—pharmacological interventions.

Ref.	No. Subjects Age (years)	Inclusion Criteria	Design	Type of Intervention	Intervention Effect	Notes
[42]	16 post-menopausal women Mean age, 55	BMI ≥ 25kg/m^2^	Cross-over PL-controlled RCT 3-month washout in-between	HRT (12 weeks) Placebo (PL) (12 weeks) No diet or exercise advice	FFM (kg): HRT + 0.35, *p* < 0.05 vs. PL; PL: −1.0, *p* < 0.05 vs. pre-treatment Total bone mineral density (g/cm^2^): HRT +8.6, *p* < 0.05 vs. PL; PL−3.9 *p* < 0.05 vs. BL Abdominal fat mass (kg): HRT −0.19, *p* < 0.05 vs. BL; PL +0.25 *p* < 0.05 vs. BL	Mean BMI = 27 kg/m^2^ Mean BF = 43% Body composition measured by DEXA
[43]	Subjects (62 men, 66 women) pre-sarcopenic and deficient in vitamin D w/wo associated obesity Mean age, 73	Presarcopenia as skeletal muscle mass/height^2^ <5.45 kg/m^2^ for women Serum level of 25(OH)D < 20 ng/mL Obesity defined as BMI ≥ 30 kg/m^2^	Parallel group (*n* = 2) controlled RCT Duration, 6 months	10,000 IU cholecalciferol 3/week (vitamin D) Placebo (PL)	HS: no difference in vitamin D vs. PL Appendicular skeletal muscle mass (ASMM): increased in vitamin D group (*p* < 0.001 vs. PL); (two-way ANOVA) effect of vitamin D on ASMM much higher in non-obese vs. obese subjects. (1.57 vs. 1.32, *p* < 0.001). No sex-related effect was observed in the presarcopenic obese group	Obesity in 49% of study population Body composition measured by DEXA

AE—aerobic exercise; BF-body fat; BL—baseline; BMI—body mass index; BF—body fat; DEXA—dual-energy x-ray absorptiometry; FFM—fat-free mass; HGS—handgrip strength; HRT—hormone replacement therapy; RCT—randomized controlled trial.

**Table 3 nutrients-11-01302-t003:** Prevention of Sarcopenic Obesity—exercise and physical therapy.

Ref.	No. Subjects	Inclusion Criteria	Design	Type of Intervention	Intervention Effect	Notes
[44]	62 sedentary women aged ≥60 Mean age, 67	Physical independency Obesity not mentioned	Parallel groups (*n* = 3) RCT Duration,12 weeks	RT 1 set (30 min) 3/week (GS1) RT 3 sets (50 min) 3/week (GS3) Control (no exercise − C)	Strength (%): GS1 + 18.5, GS3 + 25, both *p* < 0.05 vs. C (−7.2); GS3 *p* < 0.05 vs. GS1 SMM (kg): GS1 +0.9, GS3 +1.1, both *p* < 0.05 vs. C (+0.2) Body fat (%): GS1: −0.4, GS3 −2.5, both *p* < 0.05 vs. C (+0.6); GS3 *p* < 0.05 vs. GS1	Mean BMI = 27 kg/m^2^ Body composition measured by DEXA
[45]	41 sedentary obese non- sarcopenic women aged ≥ 60 years Mean age, 66	Body fat > 32% AFFM above a population specific cut-off	Parallel groups (*n* = 2) RCT comparing non-sarcopenic with SO Duration 16 weeks	RT (2 sessions of 40–50 min/week) All subjects advised not to change usual diet	In the subgroup of non-sarcopenic obese: BF: −0,6 kg, *p* = 0,03 vs. BL. No changes in AFFM vs. BL 30 s chair stand-up and timed-up-and-go: improved vs. BL (*p* = 0.000) Strength parameters: improved (moderate effect size) vs. BL (*p* ≤ 0.01)	Mean BMI 28 kg/m^2^ Body composition measured by DEXA
[46]	74 women out of 136 abdominally obese adults aged 60–80 Mean age, 67	WC ≥ 88 cm in women No conditions incompatible with exercise engagement	Parallel groups (*n* = 2) RCT Duration 6 months	Control, no exercise AE (150 min/week) RT (60 min/week) AE + RT (150 min/week) All on isocaloric diet	Combined *z*-score of tests for functional limitation improved AE, RT, RT + AE *p* < 0.05 vs. control; RT + AE *p* < 0.05 vs. AE and RT Oxygen consumption (peak VO2) increased in AE and RT + AE vs. RT (*p* < 0.05) and vs. control (*p* < 0.05) Skeletal muscle increased in RT and RT + AE vs. AE (*p* < 0.05) and vs. control (*p* < 0.05)	Mean BMI = 30 kg/m^2^ Women-specific data not provided (responses not different between sexes within treatment groups) Body composition measured by MRI

AE—aerobic exercise; AFFM—appendicular fat-free mass; BL—baseline; BMI—body mass index; BF—body fat; DEXA—dual-energy x-ray absorptiometry; MRI—magnetic resonance imaging; RCT—randomized controlled trial; SMM—skeletal muscle mass; WC—waist circumference.

**Table 4 nutrients-11-01302-t004:** Prevention of Sarcopenic Obesity—combined interventions.

Ref.	No. Subjects Age (years)	Inclusion Criteria	Design	Type of Intervention	Intervention Effect	Notes
[47]	439 overweight or obese post- menopausal sedentary women Mean age, 58	BMI ≥ 25.0 kg/m^2^ (≥23.0 if Asian American) Sarcopenia as SMI ≤ 5.67 kg/m^2^	Parallel groups (*n* = 4) RCT Duration 12 months	Moderately hypocaloric diet (D) Exercise (AE − 225 min/week) Combined (D + AE) Control (C-no intervention)	Total FFM (kg): D: −1.1 vs. −0.1 C, *p* < 0.01; AE: no significant changes; D + AE: −0.6 kg, *p* > 0.01 vs. AE Appendicular FFM (kg): D −0.5, *p* = 0.02 vs. control; AE 0.0 kg (not significant vs. C), D + AE − 0.2, (*p* < 0.01 vs. D) No differences in sarcopenia incidence among non-sarcopenic women	17% at BL (mean BMI 31 kg/m^2^) had sarcopenia Body composition measured by DEXA
[48]	31 overweight or obese, postmenopausal women Mean age, 65	BMI ≥ 28 kg/m^2^	parallel group (*n* = 2) RCT Duration 6 months	Hypocaloric diet + whey protein (2 × 25 g/day) (PRO) Hypocaloric diet supplemented with maltodextrine (CARB) Mild exercise (flexibility + aerobic 40′–50′ sessions 2–3/week) in both groups	Whole body mass (kg): CARB −3.6, PRO −7.7; (*p* = 0.051 vs. CARB) Thigh muscle mass (%): CHO +4.5, PRO +10.3 (*p* = 0.049 vs. CARB) Intermuscular adipose tissue (cm^2^): CARB −1.0, PRO −9.2; (*p* = 0.03 vs. CARB) No differences in changes in strength, balance, or physical performance measures between PRO and CARB	Mean BMI = 33.4 kg/m^2^ Body composition measured by DEXA and MRI
[49]	54 overweight and obese sedentary women aged 60–75 Mean age, 66	BMI ≥ 27 kg/m^2^ and/or body fat percentage above 35%	Parallel groups (*n* = 3) RCT Duration 14 weeks	Exercise (RT, 3/week), no diet (Ex) Ex + low-calorie high-CHO diet (ExHC) Ex + low-calorie high-protein diet (ExHP)	No reduction in FFM in all groups Percent BF: Ex −2.0%; ExHC −4.3; ExHP −6.3%; *p* = 0.002 vs. Ex and HP Strength increased in all groups with no group interaction	Mean BMI 30 kg/m^2^ Target HC diet: 55% HC, 15% P, 30% fat Target HP diet: P 1.2 g/kg, 30% fat Body composition by DEXA
[50]	94 post-menopausal sedentary women Age range 40–65	Being either on HRT (*n* = 39) or not HRT (*n* = 55)	Parallel groups (*n* = 4) RCT (2 × 2 factorial design) Duration, 12 months	Exercise (RT + weight bearing training) 3/week + HRT Exercise, no HRT No exercise, HRT No exercise, no HRT	Exercise groups: FFM total (+12%), arm (+15%), leg (+11%; strength (+9–20%); % BF (−1.9%) vs. BL (*p* < 0.001); No significant differences in no-exercise groups No interaction effects of HRT	Mean BF = 38% at BL Body composition measured by DEXA
[51]	40 women and 48 men nondiabetic overweight/obese aged 65–79 Mean age, 70	SPPB 3–10 (values < 10 predictive of mobility and disability risk)	Parallel groups RCT (*n* = 4) with 2 × 2 factorial design Duration 16 weeks	Hypocaloric diet (D) + Resistance Training (RT) D + RT + pioglitazone 30 mg (PIO) D + PIO D only	Women overall: WL −6.5%; FM −9.7%; LM –4.1% (all *p* < 0.05 vs. BL) Thigh muscle volume (cm^3^): RT −34 vs. no−RT 59, *p* = 0.040 Thigh subcutaneous fat (cm^3^): PIO −104 vs. no-PIO−298; mean difference 194, *p* = 0.002	Women BMI = 33 kg/m^2^ Unlike women, PIO significantly reduced abdominal fat in men Body composition by DEXA and CT

AE—aerobic exercise; BL—baseline; BMI—body mass index; BF—body fat; DEXA—dual-energy x-ray absorptiometry; FFM—fat-free mass; HRT—hormone replacement therapy; RCT—randomized controlled trial; RT—resistance training.

**Table 5 nutrients-11-01302-t005:** Treatment of Sarcopenic Obesity—Nutritional (diet and/or supplements).

Ref.	No. Subjects Age (years)	SO Definition	Design	Type of Intervention	Intervention Effect	Notes
[53]	Analysis by sex of 23 women and 17 men Mean age, 76	Sarcopenia diagnosed by the residual method Overweight or obesity were not inclusion criteria High prevalence with cases with elevated %BF	Parallel group (*n* = 2) RCT Duration 12 weeks	Intervention: Protein supplements (210 g/day of ricotta cheese) plus the habitual diet Control: habitual diet	No significant effect of protein supplementation on ASMM or strength in both sexes	Mean BF in women 41% Body composition measured by DEXA
[54]	104 women aged > 65 years with SO Mean age, 66	BMI ≥ 30.0 kg/m^2^, or WC > 88.0 cm or FM% ≥ 35.0%, or FM index ≥ 9.5 kg/m^2^ Sarcopenia defined by MM index, MM/height^2^ (kg/m^2^), as <2SD the obesity-derived cut-off score (7.3 kg/m^2^-class 2)	Parallel group (*n* = 2) RCT Duration 12 weeks	High protein (1.2 g/kg) low-calorie diet (HP) Normal protein (0.8 g/kg bw reference) low-calorie diet (NP)	BMI (kg/m^2^): NP −1.3; HP −0.8, both *p* < 0.001 vs. BL MM index (kg/m^2^): NP −0.2, *p* < 0.01 vs. BL; HP +0.2, *p* < 0.01 vs. BL Arm-muscle area (cm^2^): NP −5.7, *p* < 0.001 vs. BL; HP −0.5, n.s. No significant difference in HGS vs. BL	Mean BMI = 31.5 kg/m^2^ Body composition measured by BIA and anthropometry
[55]	18 women aged 41–74 years with SO Mean age, 55	Obesity defined FM >34.8%; Sarcopenia defined by lean body mass <90% of the subject’s ideal FFM	Parallel group (*n* = 2) RCT (pilot)	Low-calorie high-protein diet (1.2–1.4 g/ kg bw reference/day) (HP) Low-calorie diet plus placebo (control)	WL: HP −3.9 kg (*p* = 0.01 vs. BL); control −3.8 kg (*p* = 0.05 vs. BL) FFM: HP +2.3 kg (*p* = 0.05 vs. control); control +0.6 kg (n.s) FM: HP −9.7 kg (*p* = 0.01 vs. BL); control −7.3 kg (*p* = 0.03 vs. BL) HGS: HP +1.6 kg (*p* = 0.01 vs. BL); control: n.s. No significant change in SPPB for both groups	Body composition measured by BIA

ASMM—appendicular skeletal muscle mass; BIA—bioelectrical impedance analysis; BL—baseline; BF—body fat; BMI—body mass index; DEXA—dual-energy x-ray absorptiometry; FFM—fat-free mass; FM—fat mass; HGS—handgrip strength; MM—muscle mass; PL—placebo; RCT—randomized controlled trial; SPPB—short physical performance battery; WC—waist circumference; WL—weight loss.

**Table 6 nutrients-11-01302-t006:** Treatment of sarcopenic obesity—Pharmacological interventions.

Ref.	No. Subjects Age (years)	SO Definition	Design	Type of Intervention	Main Intervention Effect	Notes
[56]	18 post- menopausal women with SO aged 50–70 Mean age, 58	Muscle mass (MM) index <6.87 kg Appendicular FFM/m^2^ FM > 40%	Parallel group (*n* = 2) PL-controlled RCT Duration 6 months	Isoflavones 70 mg (ISO) (*n* = 12) Placebo (PL) (*n* = 6)	Leg FFM (kg): ISO +0.29 vs. PL −0.62, *p* = 0.034 Appendicular FFM (kg): ISO +0.53, PL −0.78, *p* = 0.016 MM index: ISO +0.26, PL−0.27, *p* = 0.037	BMI = 29 kg/m^2^ Body composition by DEXA

BMI—body mass index; DEXA—dual-energy x-ray absorptiometry; FFM—fat-free mass; FM—fat mass; MM—muscle mass; PL—placebo; RCT—randomized controlled trial.

**Table 7 nutrients-11-01302-t007:** Treatment of sarcopenic obesity—Exercise and physical therapy.

Ref.	No. Subjects Age (years)	SO Definition	Design	Type of Intervention	Intervention Effect (Main Findings)	Notes
[57]	113 overweight and obese elderly women Mean age, 67	BMI ≥ 25 kg/m^2^ appendicular FFM by residual values method including height and FM	Parallel groups (*n* = 2) RCT	Resistance exercise (RE) 3/week Control (C − no exercise) Duration 24 weeks	Total FFM (kg): RE: +0.6; *p* < 0.01 vs. BL and vs. C Appendicular FFM (kg): RE: +0.29; *p* < 0.01 vs. BL and vs. control Strength (Isokinetic relative peak torque 60°) (Nm/kg × 100) RE: + 20.6; *p* < 0.01 vs. BL and vs. C	BMI (27.1–29.1 kg/m^2^) Body composition measured by DEXA
[58]	60 sarcopenic overweight and obese elderly (83% women) Mean age, 69	BMI ≥ 25 kg/m^2^ and visceral fat area ≥ 100 cm plus skeletal MM ≤ 25.7% b.w.	Parallel groups (*n* = 4) RCT	Resistance/Aerobic Exercise (RT or AE) Combination (AE + RT) Control (C − no exercise) All sessions 2/week Duration 8 weeks	HGS (kg): RT: +3.5, *p* < 0.05 vs. all other groups, no changes in AE and RT + AE Skeletal MM (kg): RT: +0.1, AE: +0.1, RT+AE: +0.2 (in all *p* < 0.05 vs. C) FM (kg): RT: −1., AE: −0.7, RT + AE: −1.1 (in all *p* < 0.05 vs. C) Back extensor (kg): RT: +9.0, AE: +7.9, RT + AE: + 10.0 (in all *p* < 0.05 vs. C)	BMI (26.8–29.0kg/m^2^) Body composition measured by BIA Effect persisted 4 weeks after end of intervention
[45]	8 sedentary women with obesity aged ≥ 60 years Mean age, 66	body fat % > 32 Appendicular fat-free mass less than population- specific cut-off	Parallel group (*n* = 2) RCT of women w/wo SO Duration 16 weeks	RT (2 sessions of 40–50 min/week)	In the subgroup of women with SO: no difference in %BF, 30 s chair stand-up, timed-up-and-go vs. BL Improved strength vs. BL (*p* ≤ 0.01) with trivial effect sizes	Mean BMI = 28 kg/m^2^ Body composition measured by DEXA
[59]	50 women aged ≥ 65 years with SO Mean age, 74	BMI ≥ 25.0 kg/m^2^ + ASMM/weight < 25.1 %	Parallel groups (*n* = 2) RCT Duration 24 weeks	Combined RT and AE 5/week (Ex) Control (C − no exercise)	BF (%): Ex −2.0, *p* < 0.01 vs. BL; C: n.s. No effect on appendicular lean mass. HGS (kg): Ex +2.5, *p* < 0.001 vs. BL and vs. C; C −0.5, *p* < 0.05 vs. BL Maximum walking speed (m/s): Ex +0.15, *p* < 0.01 vs. BL and vs. C; C −0.04, *p* < 0.01 vs. BL	Body composition by BIA Improvement in carotid artery IMT and flow velocity
[60]	35 women aged 60–80 years with SO Mean age, 67	BF > 30% SMI <7.15 kg/m^2^	Parallel groups (*n* = 2) RCT Duration 12 weeks (intervention) + follow- up at 9 months	Elastic band resistance training (RT) 3 times/week Control (C-no exercise)	Results are reported at 9-mo follow-up. Absolute muscle mass: RT +0.72 kg, *p* < 0.01 vs. C); similar results for appendicular lean mass and SMI Global physical capacity score: RT + 4.22, *p* < 0.001 vs. C). Clinically significant improvement in all functional tests. Physical component score (SF-36): RT +15.06, *p* < 0.001 vs. C)	Mean BMI = 28 kg/m^2^ Body composition measured by BIA
[61]	35 women aged ≥ 60 years with SO Mean age, 69	BF > 30% SMI <27.6%	Parallel groups (*n* = 2) RCT Duration 12 weeks	Elastic band resistance training (RT) 3 times/week Control (home exercise)	Total BF: RT −0.58 kg, *p* = 0.03; control: n.s. Total bone density: RT +0.06 g/cm^2^, *p* = 0.026; control: n.s. No effect on lean appendicular mass	Mean BMI = 28 kg/m^2^ Body composition measured by BIA (screening) and DEXA (treatment)
[62]	17 SO subjects (95% women) aged ≥ 60 years Mean age, 71	BMI > 30 kg/m^2^ plus EWGSOP1 criteria	Parallel groups (*n* = 2) RCT	High-speed power training circuit (HSC) Standard strength hypertrophy training (ST) 2-week adaptation before treatment Duration 15 weeks	HSC improved physical function (SPPB) by 20% (_adj_mean difference 1.1; *p* = 0.02, effect size g = 0.6 with no changes in ST group No change for SMI, BF %, 6MWT, HGS vs. BL in both groups. Few IADL tasks with negligible to small changes for either HSC or ST	Adherence rates > 80% Lower ratings of perceived exertion in HSC vs. ST Subjects in ST with mild to moderate acute joint pain.

AE—aerobic exercise; ASMM—appendicular skeletal muscle mass; BF—body fat; BIA—bioelectrical impedance analysis; BL—baseline; BMI—body mass index; DEXA—dual-energy x-ray absorptiometry; EWGSOP1—European Working Group on Sarcopenia in Older People 1 (2010 criteria); FFM—fat free mass; FM—fat mass; HGS—handgrip strength; IADL—Instrumental Activities of Daily Living; IMT—intima-media thickness; MM—muscle mass; 6MWT—6-min walking test; RT—resistance training; RCT—randomized controlled trial; SF-36—Short-Form 36 Questionnaire; SMI—skeletal muscle index.

**Table 8 nutrients-11-01302-t008:** Treatment of sarcopenic obesity—Combined interventions.

Ref.	No. Subjects Age (years)	SO Definition	Design	Type of Intervention	Intervention Effect	Notes
[47] *see also Table 4*	Subgroup of 76 post menopausal sedentary women with SO Mean age, 58	BMI ≥ 25.0 (or ≥23.0 kg/m^2^ if Asian American) Sarcopenia defined as SMI ≤ 5.67 kg/m^2^	Parallel groups (*n* = 4) RCT Duration 12 months	Moderately hypocaloric diet (D) Aerobic exercise (AE) Combined D +AE Control (C-no intervention)	14% in C, 8% in D, 50% in AE, 35% in the D + AE no longer met the sarcopenia criteria by 12 months. No subgroup-specific statistical analysis was provided.	17% with sarcopenia (mean BMI = 31 kg/m^2^) Body composition measured by DEXA
[63]	75 women aged ≥ 60 years with SO Mean age, 77	Obesity as > 35% BF Sarcopenia as SMI < 5.75 kg/m^2^	Parallel groups (*n* = 3) RCT Duration 26 weeks	Whole-body electro- myostimulation (WB-EMS, 1/week) WB-EMS + protein + vitamin D supplements (WB-EMS&P) Non-training controls (C)	SMI (kg/m^2^): WB-EMS +0.14, WB-EMS&P + 0.11; both *p* < 0.001 vs. C. Gait speed: WB-EMS 0.08 m/s, *p* = 0.026 vs. C; WB-EMS&P n.s. No significant changes in BF or HGS in all treatment groups	Mean body fat = 37% All groups were supplemented with vitamin D Body composition by DEXA
[64]	139 women aged ≥ 70 years with SO Mean age, 81	BF ≥32% and SMI < 5.67 kg/m^2^ or HGS < 17.0 kg or walking speed < 1.0 m/s.	Parallel groups (*n* = 4) RCT Duration 12 weeks	Exercise (RT + AE − 2/week) + EAA (3 g) + catechins + vitamin D) (ExNu) Exercise only (Ex) Nutritional intervention only (N) Control (health education)	Body FM decreased significantly in all groups vs. BL; ExNU −1.0 kg (*p* = 0.036 vs. N) Step length: ExNu +3.2 cm; Ex +3.5 cm (*p* = 0.007 vs. N); both significantly increased vs. BL No significant changes in SMI and HGS among groups	Body composition measured by DEXA (screening) and BIA (treatment)

AE—aerobic exercise; BIA—bioelectrical impedance analysis; BL—baseline; BF—body fat; BMI—body mass index; DEXA—dual-energy x-ray absorptiometry; EAA, essential amino acids; FM—fat mass; HGS—handgrip strength; RT—resistance training; RCT—randomized controlled trial; SMI—skeletal muscle index; WB-EMS—whole-body electromyostimulation.

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
