# Peer review of "Prevention and Treatment of Sarcopenic Obesity in Women"

_nutrients, 2019, doi:10.3390/nu11061302_

Round 1

Reviewer 1 Report

To the Authors,

Petroni and colleagues provide an interesting and insightful review of the literature examining the efficacy of exercise, nutrition, pharmacology and combined strategies in the prevention of sarcopenic obesity in older women and treatment in women classified as sarcopenic obese. Whilst the review is somewhat similar to what has recently been published in Nutrients, this review provides a novel and direct focus on pre-sarcopenic obese and sarcopenic obese women. The tables detailing the information from their literature search is detailed and thorough and provides a good overview of the literature. I do have some suggestions below on ways to improve what is already a good manuscript which I have enjoyed reading.

Major comments

A remarkably similar review has already been published in Nutrients (Trouwborst et al., 2018; 10(5): 605), where the first three chapters are similar in terms of layout and content. A simple Google search revealed this article, yet the authors have not recognised it within their publication.

As with Trouwborst et al. (2018), the authors require a section within the manuscript, perhaps after the definition of sarcopenic obesity, outlining the aetiology of sarcopenic obesity in women, outlining changes in body composition, skeletal muscle contractile function, and the overall biomechanical implications of sarcopenic obesity. Whilst the authors briefly describe the functional limitations in sarcopenic adults on lines 64-69 and lines 139-141, a more in-depth discussion of the implications of sarcopenic obesity beyond what is currently described is required. The proposed section could outline the muscle and adipose tissue size, and then outline changes in the force generating capacity of muscles in absolute terms, describe the changes in muscle quality (force or power normalised to muscle size) and the overall biomechanical influences and alterations in activities of daily living when sarcopenic obese women are compared against old obese and old sarcopenic women (and perhaps even against men to better drive home the fact women are at greater risk and therefore requires the most focus). Data exists for these measures (see reviews by Tomlinson et al., 2016; Biogerontology, 17(3):467-83. & Tallis et al., 2018; J Exp Biol, 221, Pt 13). By providing an indication of changes in muscular function and how that influences older women, the interventions will be better contextualised as a discussion of the aforementioned factors will have been provided to the reader. Moreover, by describing changes in muscle quality, where measures of muscle size have been performed using imaging techniques described in this paper (ie DEXA, CT), a link between imaging techniques in section three and muscle function can be made since the descriptions are already in place.

The opening paragraph (lines 40-42) would benefit from some demographics and perhaps health-related/financial implications to better contextualise why sarcopenic obesity is a cause for concern.

As a lay reader, I would probably still be none the wiser as to what the best strategy should be used to prevent or treat sarcopenic obesity. Somewhere in the discussion, and perhaps related to figure 1 (see below comment), an explicit outline of what is recommended to prevent and treat sarcopenic obesity in women is required. This can be focused from a change in body composition or contractile function or even an overall improvement in quality of life.

Figure 1 – To me, without a more detailed figure legend, this figure is not very clear. Is the triangle meant to represent the most effective strategies at the base and least effective at the top? Or is the colour transition meant to represent strategy efficacy? Or is the base meant to represent a greater abundance of evidence for diet and nutrition and the tip least evidence for pharmacological interventions, like what a food pyramid shows? I would recommend a complete overhaul of figure 1. Perhaps two triangles could be used, one for prevention and another for treatment, or a colour-coded figure which highlights the most effective and least effective strategies (e.g. green for strong evidence, amber for cautious/equivocal evidence and red for weak evidence).

Within section 6.3, a comment on which sex responds best to strategies targeting sarcopenic obesity would be beneficial. This could be described as a change in muscle morphology, contractile function or performance of activities of daily living. Whilst pharmacotherapy is very likely to be sex-specific, I wonder whether females respond better to certain strategies better than females and vice versa. This will provide a nice basis for further study and provide direction for geriatricians and researchers to better determine which strategies have the best outcome and whether this is sex-specific.

Minor

Line 41 - A space is required between reference one (i.e. [1].) and the new sentence.

Instances where BMI is expressed in kilograms per meter squared, or kg/m2 should have a superscript number two. This pertains to lines 109, 115 and 364, and notes for Ref. 24 in table 1A

As with BMI, muscle density and CSA require a superscript for the number two. This pertains to g/cm2 for intervention effect of Ref. 27 of table 1B, line 304, g/cm2 for intervention effect of Ref. 46 of table 2C,

Line 115 – change “this latter” to “the latter”.

Line 403 – Full stop required after reference 51.

Line 420-421 – The end of the sentence regarding hypocaloric diets seems incomplete.

To improve clarity, I suggest that the date ranges of the included studies could be provided in section 2.

Author Response

Prevention and treatment of sarcopenic obesity in women

Maria L Petroni,1 Maria T Caletti,2 Riccardo Dalle Grave,3 Alberto Bazzocchi,4 Maria P A Gómez, 5,6 and Giulio Marchesini7

Point-by-point response to reviewers’ criticism

REVIEWER #1

We are very grateful to the Reviewer for his/her remarks and detailed suggestions, that we have implemented in the revised version of the paper. We believe that – thank to the suggestions provided – the review has significantly improved

Petroni and colleagues provide an interesting and insightful review of the literature examining the efficacy of exercise, nutrition, pharmacology and combined strategies in the prevention of sarcopenic obesity in older women and treatment in women classified as sarcopenic obese. Whilst the review is somewhat similar to what has recently been published in Nutrients, this review provides a novel and direct focus on pre-sarcopenic obese and sarcopenic obese women. The tables detailing the information from their literature search is detailed and thorough and provides a good overview of the literature. I do have some suggestions below on ways to improve what is already a good manuscript which I have enjoyed reading.

·        We thank the reviewer for his/her positive comments.

Major comments

A remarkably similar review has already been published in Nutrients (Trouwborst et al., 2018; 10(5): 605), where the first three chapters are similar in terms of layout and content. A simple Google search revealed this article, yet the authors have not recognised it within their publication.

·       We have added a new paragraph in the Introduction (lines 58-63) where we acknowledged the previous narrative review of Trouwborst and coll. and explain the reasons for writing the present review.

  As with Trouwborst et al. (2018), the authors require a section within the manuscript, perhaps after the definition of sarcopenic obesity, outlining the aetiology of sarcopenic obesity in women, outlining changes in body composition, skeletal muscle contractile function, and the overall biomechanical implications of sarcopenic obesity. Whilst the authors briefly describe the functional limitations in sarcopenic adults on lines 64-69 and lines 139-141, a more in-depth discussion of the implications of sarcopenic obesity beyond what is currently described is required. The proposed section could outline the muscle and adipose tissue size, and then outline changes in the force generating capacity of muscles in absolute terms, describe the changes in muscle quality (force or power normalised to muscle size) and the overall biomechanical influences and alterations in activities of daily living when sarcopenic obese women are compared against old obese and old sarcopenic women (and perhaps even against men to better drive home the fact women are at greater risk and therefore requires the most focus). Data exists for these measures (see reviews by Tomlinson et al., 2016; Biogerontology, 17(3):467-83. & Tallis et al., 2018; J Exp Biol, 221, Pt 13). By providing an indication of changes in muscular function and how that influences older women, the interventions will be better contextualised as a discussion of the aforementioned factors will have been provided to the reader. Moreover, by describing changes in muscle quality, where measures of muscle size have been performed using imaging techniques described in this paper (ie DEXA, CT), a link between imaging techniques in section three and muscle function can be made since the descriptions are already in place.

·       A new section (1.1 Age-related and obesity-related changes in muscle composition, structure and function in women) lines 90-106 has been added to outline changes in body composition, skeletal muscle contractile function, and the overall biomechanical implications of sarcopenic obesity. Imaging techniques used for documenting anatomical changes have also been reported.

The opening paragraph (lines 40-42) would benefit from some demographics and perhaps health-related/financial implications to better contextualise why sarcopenic obesity is a cause for concern.

·       Demographics and health-related/financial implications of sarcopenic obesity have been added (lines 40-45 and 49-50) to better contextualise why sarcopenic obesity is a cause for concern.

As a lay reader, I would probably still be none the wiser as to what the best strategy should be used to prevent or treat sarcopenic obesity. Somewhere in the discussion, and perhaps related to figure 1 (see below comment), an explicit outline of what is recommended to prevent and treat sarcopenic obesity in women is required. This can be focused from a change in body composition or contractile function or even an overall improvement in quality of life.

Figure 1 – To me, without a more detailed figure legend, this figure is not very clear. Is the triangle meant to represent the most effective strategies at the base and least effective at the top? Or is the colour transition meant to represent strategy efficacy? Or is the base meant to represent a greater abundance of evidence for diet and nutrition and the tip least evidence for pharmacological interventions, like what a food pyramid shows? I would recommend a complete overhaul of figure 1. Perhaps two triangles could be used, one for prevention and another for treatment, or a colour-coded figure which highlights the most effective and least effective strategies (e.g. green for strong evidence, amber for cautious/equivocal evidence and red for weak evidence).

·       Based on Reviewers’ comments we have re-drawn the Figure in order to make it simpler to understand. Now there are three circles in part overlapping. There is no distinction between prevention and treatment since the possible interventions are overall quite similar. The front circle represents physical activity, the intermediate circle represents nutrition, while the back circle represents pharmacological treatment. We also provided more explanation in the Figure legend (lines 479-485) about best strategies to prevent or treat sarcopenic obesity. We hope this will be sufficiently clear to the lay reader.

Within section 6.3, a comment on which sex responds best to strategies targeting sarcopenic obesity would be beneficial. This could be described as a change in muscle morphology, contractile function or performance of activities of daily living. Whilst pharmacotherapy is very likely to be sex-specific, I wonder whether females respond better to certain strategies better than females and vice versa. This will provide a nice basis for further study and provide direction for geriatricians and researchers to better determine which strategies have the best outcome and whether this is sex-specific.

·       In lines 554-566, as suggested by the Reviewer a couple of paragraphs have been added on gender-related effect by strategies targeting sarcopenic obesity.

Minor

Line 41 - A space is required between reference one (i.e. [1].) and the new sentence. Done

Instances where BMI is expressed in kilograms per meter squared, or kg/m2 should have a superscript number two. This pertains to lines 109, 115 and 364, and notes for Ref. 24 in table 1° Done

As with BMI, muscle density and CSA require a superscript for the number two. This pertains to g/cm2 for intervention effect of Ref. 27 of table 1B, line 304, g/cm2 for intervention effect of Ref. 46 of table 2C, Done

Line 115 – change “this latter” to “the latter”. Done

Line 403 – Full stop required after reference 51. Done

Line 420-421 – The end of the sentence regarding hypocaloric diets seems incomplete. Done

To improve clarity, I suggest that the date ranges of the included studies could be provided in section 2. Done

·       Typos highlighted by the Reviewer have been corrected. Also, an English mother tongue reader for grammatical errors and style improvement has revised the manuscript.

Reviewer 2 Report

In their study, Petroni et al. aimed to elucidate prevention and treatment options of sarcopenic obesity (SO). 

For this purpose, they reviewed recent approaches pertaining sarcopenic obesity in women. However, the impact of their conclusions is limited since only a short period (jan 29th- march 14th) has been covered and the narrative review does not contain meta-analyses.

Even though a contemporary issue, latest definition of sarcopenia has not been taken as basis.

Additionally, we recommend the PRISMA statement for constitution of a review (Liberati et al. 2009).

Introduction & Definition:
Regarding sarcopenia, please refer to Cruz-Jentoft et al. 2019, especially definition criteria have changed, e.g. grip strength <27kg (men), <16kg (women), ...

If this short timeline of including studies, consider using more than Medline and Google Scholar (e.g. Cochrane database, Embase, ...).

4.1 The little number of studies limits the significance of the assumptions. Please give numbers of examined patients.

4.2.2: Could not find the heading.

Discussion

Figure 1: Please rearrange and give a less chaotic design.

The figure legends should be more detailed. They should include the n-numbers and briefly describe the meaning/consequence of the results/comparisons.

The manuscript has some typing and grammatical errors which need to be rectified.

Author Response

REVIEWER #2

In their study, Petroni et al. aimed to elucidate prevention and treatment options of sarcopenic obesity (SO). For this purpose, they reviewed recent approaches pertaining sarcopenic obesity in women. However, the impact of their conclusions is limited since only a short period (jan 29th- march 14th) has been covered and the narrative review does not contain meta-analyses. Even though a contemporary issue, latest definition of sarcopenia has not been taken as basis. Additionally, we recommend the PRISMA statement for constitution of a review (Liberati et al. 2009). If this short timeline of including studies, consider using more than Medline and Google Scholar (e.g. Cochrane database, Embase, ...).

·       We thank the Reviewer for his/her useful comments and suggestions. The main criticism raised was that this is a narrative review with a supposed low level of evidence. The crucial issue is that clinical intervention studies on sarcopenic obesity prevention and treatment are widely heterogeneous in terms of definition of disease, methodologies employed for diagnosis, study design and outcome measures.

·       This was an invited review for a Special Issue of Nutrients and we were given only a few weeks’ time to complete it. Ordinarily, a systematic review requires between 18 to 24 months to be completed, so this was clearly not feasible. Moreover, no metanalysis on the topic has been published to our knowledge.  Even if we did a systematic review, the chance to produce strong evidences on this debated topic are at present quite poor. We discuss this in detail in lines 52-58. Nevertheless we believe that a narrative review under the current setting makes sense since can provide readers with an overview on state-of-the-art about studies on prevention and treatment of SO in middle-aged and older-aged women and to highlight new research areas not yet addressed. We also discuss this in lines 58-66.

·       Being a narrative review the process of acquisition and management of data does not fit into the PRISMA statement that is recommended for systematic reviews and metanalysis. However, in order to provide as much reproducibility as possible and to show how we tried to decrease/avoid selection bias, we further extended Section 2. We stated that “further publications of potential interest were identified as citations in the articles retrieved during the first search” (lines 139-140), and “all selected studies could be retrieved as full papers” (lines 147-148).

·       Moreover we were more explicit about the timer range of studies reported. Lines 154-156 now quote: “The publication date of the included studies ranges between 2001 and 2019. Review includes 24 papers including 1820 women (90%) out of total number of 2014 enrolled subjects”.

Introduction & Definition:

Regarding sarcopenia, please refer to Cruz-Jentoft et al. 2019, especially definition criteria have changed, e.g. grip strength <27kg (men), <16kg (women)

·       We are grateful to the Reviewer for the reporting of the updated EWGSOP criteria (EWGSOP2), which were published during the writing of the present review. We have included these criteria in the revised version of the paper (lines 193-206; Table 2c and line 453) and compared them with the earlier EWGSOP criteria (lines 203-205). We also added a remark taken from the EWGSOP2 document about sarcopenic obesity representing a distinct condition from sarcopenia (lines 207-208).

4.1 The little number of studies limits the significance of the assumptions. Please give numbers of examined patients.

·       Number of patients is included in the Tables

·       We have remarked in both the Results (lines 251, 380) and the Discussion (lines 517, 537) the small number of patients in the pharmacologicals intervention making more dubitative statements about their efficacy.

4.2.2: Could not find the heading.

·       Corrected

Discussion

Figure 1: Please rearrange and give a less chaotic design.

The figure legends should be more detailed. They should include the n-numbers and briefly describe the meaning/consequence of the results/comparisons.

·       Based on Reviewers’ comments we have re-drawn the Figure in order to make it simpler to understand. Now there are three circles in part overlapping. There is no distinction between prevention and treatment since the possible interventions are overall quite similar. The front circle represents physical activity, the intermediate circle represents nutrition, while the back circle represents pharmacological treatment. We also provided more explanation in the Figure legend (lines 479-485) about best strategies to prevent or treat sarcopenic obesity. We hope this will be sufficiently clear to the lay reader.

The manuscript has some typing and grammatical errors which need to be rectified.

·       Minor typos have been corrected. Also, the manuscript has been revised by a English mother tongue reader for grammatical errors and style improvement.

Round 2

Reviewer 1 Report

To the authors,

Thank you for systematically address my suggestions. The review has been substantially improved compared to the original revision, with figure 1 and the associated figure caption far clearer than the original submission. However, I do have a few minor suggestions which are predominantly cosmetic, meaning wholesale changes to the manuscript are not required.

Line 59
Thank you for recognizing the review by Trouwborst et al. and providing a justification for your MS. I would delete “published in Nutrients” as it is not common to state where a paper has been published in the main text.

Line 75
As the percentage values have become more negative, they can’t increase to -12% and -9%. I would rephrase to avoid confusion.

Line 96
I’d change tension to force (or power if necessary). This is because in the current state, “tension per unit of skeletal muscle” describes what specific tension is!  Therefore, force relative to muscle size is specific tension, just to be explicitly clear.

Line 105
A space is required between reference 13 and the new sentence.

Line 545
"Single muscle fibre functions" needs to be more specific. If force was unchanged or increased in men, then state force.

Line 559-560
The grammar of this sentence is a little strange. This could be changed to:
“Pharmacological agents other than estrogen have demonstrated sex-specific effects on body and muscle composition. For example, Pioglitazone resulted in significant sex-based differences in abdominal fat, where abdominal fat loss was significantly greater in men, with no change in women [46].”
However, I shall let you decide whether to adopt the above sentence or to rewrite this particular sentence.

General
I gave a presentation recently to animal biomechanists, where someone suggested that I should use sex rather than gender when discussing differences between males and females. This is because gender is a social construct (i.e. you can change your gender or identify as a having a particular gender or as having no gender; non-binary) but sex, in humans, is biological and cannot be changed unless done so medically. I have to say I agree with this sentiment. Therefore, I would suggest using sex instead of gender where gender has been mentioned.

References
In your references, from Aleman-Mateo et al. onwards, they are a bit of a mess. I am not sure whether that should be reference 48 or reference 38. The same applies to all the references thereafter.

Author Response

Again, thank you very much for the time and effort you spent for your careful revision of the manuscript.

We followed all your suggestions:

- line 59 we deleted the words "published in Nutrients"

- line 75 was rephrased as " figure drop to -12% and -9%”

- line 96: the word "force" now substitute "tension"

- line 105: corrected 

- previous line 545 now 550-553 was changed to "However, upon resumption of physical activity and caloric intake, women, did not recover strength as measured by maximum voluntary contraction, while men did [70]. In the male sex, single muscle fiber power and isometric tension were unchanged or paradoxically increased (muscle biopsies were not carried out in women)"

- previous lines 559-560 now 566-569: we were happy to change the sentence according to your suggestions 

General: we took your point about semantic differences between "gender" and "sex". We have consequently changed most terms "gender" with "sex" or "sexes". We have occasionally left the word "gender" as it is to avoid eccessive repetitions.

Reviewer 2 Report

All my concerns have been adressed.

Author Response

We are grateful to the Reviewer for the time and effort spent in reading and commenting the manuscript. That has significantly contributed to its improvement.